## Research Article

**Key words**
Bacteriocide; disinfection; *E. coli*;
microdroplets; reactive oxygen species;
*S. typhimurium*

**Author for correspondence:**
Richard N. Zare,
zare@stanford.edu

Maria T. Dulay and Jae Kyoo Lee contributed equally to this study.

CAMBRIDGE
UNIVERSITY PRESS

# Spraying Small Water Droplets Acts as a Bacteriocide

Maria T. Dulay[1], Jae Kyoo Lee[1], Alison C. Mody[1], Ramya Narasimhan[2], Denise M. Monack[2] and Richard N. Zare[1]

[1]Department of Chemistry, Stanford University, Stanford, CA 94305, USA and [2]Department of Microbiology and Immunology, Stanford University School of Medicine, Stanford, CA 94305, USA

## Abstract

Disinfectants are important for arresting the spread of pathogens in the environment. Frequently used disinfectants are often incompatible with certain surfaces, expensive and can produce hazardous by-products. We report that micron-sized water droplets can act as an effective disinfectant, which were formed by spraying pure bulk water with coaxial nebulizing airflow. Spraying for 20 min onto *Escherichia coli* and *Salmonella typhimurium* on stainless-steel discs caused inactivation of over 98% of the bacteria. Control experiments resulted in less than 10% inactivation (water stream only and gas only) and 55% inactivation with 3% hydrogen peroxide. Experiments have shown that cell death results from cell wall destruction. We suggest that the combined action of reactive oxygen species present in water droplets (but not in bulk water) along with the droplet surface charge is responsible for the observed bactericidal activity.

## Introduction

The global market for disinfectants has rapidly grown because of increasing environmental and health concerns and rising global demands for clean food and water. Standard practices that involve the use of chemical disinfectants exist to decrease and prevent the occurrence of infectious diseases caused by spreading of pathogens on environmental surfaces, reusable medical devices and food surfaces (Centre for Disease Control, 2008; Banach *et al.*, 2015). In the United States, 9.4 million cases of foodborne illness involving 31 major pathogens, including *Escherichia coli* have been reported (Scallan *et al.*, 2011), and the numbers are even more staggering in developing countries (World Health Organization, 2015). Health care-associated pathogens infect 1 of 31 patients daily in the United States (Centre for Disease Control, 2014). Proper disinfection is an integral part of controlling these outbreaks. Although current thermal- and UV-based disinfectants are effective for a broad spectrum of pathogenic cells, they have limited utility because of difficulties in operation and possible damage to surfaces (Rushdy and Othman, 2011). On the other hand, chemical disinfectants have had more widespread use with efficiency in destroying the cell walls and disrupting the metabolism of microorganisms (Russell, 2003; Virto *et al.*, 2005; Raffellini *et al.*, 2011). Of these chemical disinfectants, oxidizing agents, such as hypochlorite (bleach), are widely used but have low biodegradability, high cost and are a source of potentially hazardous by-products (McDonnell and Russell, 1999).

Water is perhaps the most essential matter for all living systems on Earth, but water can be made to possess properties that can lead to the death of microorganisms, such as bacteria. Here, we report the development of a new disinfectant, which we call AquaROS, which is generated by spraying ordinary water into micron-sized aqueous microdroplets without the need for any applied voltage or chemical additives. AquaROS can achieve >98% inactivation of *E. coli* and *Salmonella typhimurium* within 20 min of spraying. We believe that what distinguishes small water droplets from bulk water is the presence of two surface components: (1) charges from the formation of the water droplet and (2) the presence of reactive oxygen species (ROS), such as hydrogen peroxide ($H_2O_2$) molecules, hydroxide ions ($OH^-$) and hydroxyl radicals ($OH\cdot$) (Lee *et al.*, 2019). ROS are known to be effective disinfectants that can kill bacteria with high efficacy (Vatansever *et al.*, 2013; Bogdan *et al.*, 2015). The details of the killing mechanism have not been fully established, but we believe it involves the action of charged droplets (Pillet *et al.*, 2016) in combination with ROS to cause cell wall permeation and damage. Evidence for such behaviour may be found in the use of 25-nm water nanodroplets electrosprayed from a 5-kV capillary onto bacteria-covered surfaces of stainless steel and tomato (Pyrgiotakis *et al.*, 2015).

## Materials and methods

### Generation of AquaROS microdroplets

Unless otherwise noted, microdroplets were generated by flowing chromatography-grade water (Fisher Scientific, Waltham, MA) at a flow rate of 10 µl min$^{-1}$ through a fused-silica capillary (100-µm inner diameter, 350-µm outer diameter and ~30 cm length) with coaxial $N_2$ nebulizing sheath gas at constant pressure (60, 90, 120 and 180 psi) or air compressor with pressure switching between 90 and 115 psi (~30-s ramp up to 115 psi and ~3.5-min ramp down to 90 psi). A fused-silica capillary was inserted into a stainless-steel tube with ~3 mm of the capillary outlet extended past the outlet of the stainless-steel tube. The other end of each capillary was attached via a short (~1 inch) polyethylene tube to a 1-ml gastight glass syringe (Hamilton, Franklin, MA). The syringes were placed in programmable syringe pumps for continuous flow of water during the spray experiments (Harvard Apparatus, Holliston, MA).

### Spray chamber

Three spray capillaries were bundled together and inserted in a downward direction (capillary inlet facing towards chamber bottom) through the top of 4 × 6-inch plastic box with a front vertical door. The chamber was equipped with a 0.2-µm filter. A lab jack was set at the bottom of the chamber to allow for adjustments to the spray distances. Fig. 1 illustrates schematically the chamber configurations.

### Bacterial inoculum preparation

#### Test microorganisms

Two bacteria species were used for inoculation on ethanol-cleaned and autoclaved 12-mm diameter stainless-steel discs (SPI Supplies, West Chester, PA): *E. coli* (ATCC 25922, Manassas, VA) and *S. typhimurium* (provided by D.M. Monack Lab, Stanford University). *E. coli* bacteria were cultured on lysogeny broth (LB) agar plates and incubated at 37 °C for 16–18 h. *S. typhimurium* bacteria were cultured on LB agar-streptomycin plates and incubated at 30 °C for 16–18 h. For each spray experiment, the *E. coli* inoculum was prepared by transferring a single colony from a seeded plate (prepared the day before) to ~10 ml of LB broth (Sigma-Aldrich, Milwaukee, WI) and grown overnight for 16–18 h on a shaker at 30 °C. Similarly, *S. typhimurium* inoculum was prepared by transferring a single colony from a seeded plate (prepared the day before) to ~10 ml of LB broth and grown overnight for 16–18 h at 37 °C on a rotating shaker. *E. coli* and *S. typhimurium* inoculants were then prepared by re-suspending the cultures in LB broth for a final concentration of $10^8$ CFU ml$^{-1}$.

### Inoculation of stainless-steel discs with bacterial suspensions

For experiments with *E. coli*, 5 µl of the *E. coli* bacterial suspension was deposited onto a stainless-steel disc and house vacuum-dried in a desiccator for 5 min or air-dried for ~15 min at room temperature (~25 °C) in a sterile environment from convective heating of a Bunsen burner when used on the same day as *S. typhimurium*. Five microlitres of *S. typhimurium* was deposited onto a stainless-steel disc and air-dried for ~15 min at room temperature (~25 °C) in a sterile environment from convective heating of a Bunsen burner. Each disc was placed in a sterile 15-mm diameter plastic Petri dish before inoculating and drying. Once a disc surface was dry, Aqua-ROS spraying was started. Three samples were used for each AquaROS inactivation experiment.

### Recovery of Bacteria from test surface

To recover the bacteria from a disc and to stop any on-going oxidation reaction, 2 ml of sterile LB broth was added to each Petri dish. Then the Petri dish and its contents were gently agitated. The rinsate was serially diluted with LB broth and the *E. coli* dilutions were plated on LB agar and the *S. typhimurium* dilutions were plated on LB agar-streptomycin plates. When the spread-plate

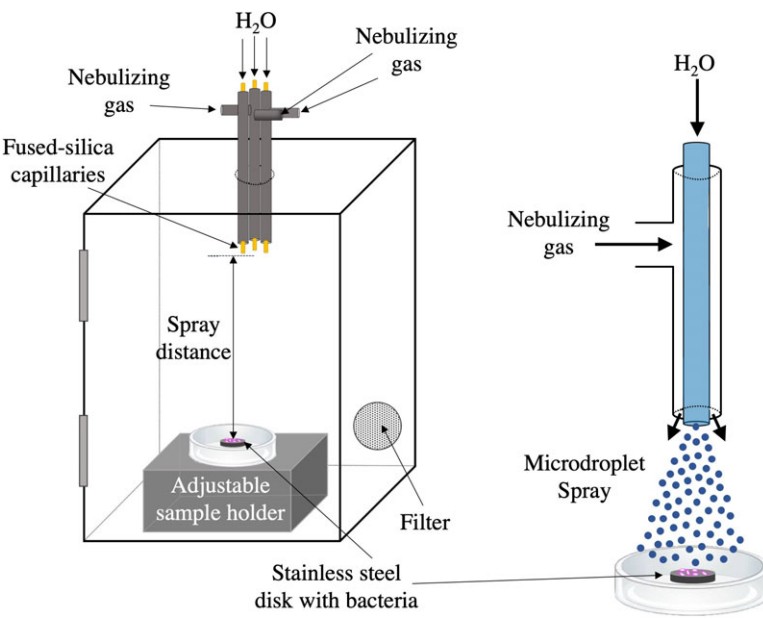

**Fig. 1.** Schematic of AquaROS disinfection device. A 4 × 6-inch plastic chamber was outfitted with a bundle of three capillary sprayers, in which each consisted of a fused-silica capillary for water flow that extended ~3 mm from the outlet of a stainless-steel tube used for the delivery of a nebulizing gas. The outlets of the capillaries were flush with each other and positioned 9 cm from the surface of the stainless-steel disc inoculated with bacteria sample. The chamber was outfitted with a 0.2-µm particulate filter. Microdroplet spray was aligned over the sample disc.

method was used, 30 µl of each dilution was added to an agar plate, and then glass beads were spread. Each sample was plated in triplicate. The spot-plate method involved the addition of 5 µl of each dilution to an agar plate (done in triplicates). The plates were incubated for 16–18 h at 37 °C for *E. coli* and 30 °C for *S. typhimurium* before colony counting.

### Bacteria inactivation experiments

#### Stainless-steel discs
The inoculated discs were placed in the spray chamber 9 cm from the capillary inlet unless otherwise noted. The alignment of the AquaROS spray was achieved visually by turning on the spray and allowing the microdroplets to accumulate onto the lid of the Petri dish containing the disc sample. The dish was adjusted manually to ensure that the disc and the accumulated droplets were vertically aligned and stabilized before removing the Petri dish lid to start the spray experiment for 20 min. For each different bacterial evaluation experiment, three discs were used to test for inactivation efficacy. Each disc was prepared prior to each spray to minimize inactivation due to drying in air. Control treatment samples were prepared in a similar manner but without exposure to AquaROS spray.

#### Bacteria inactivation analysis
All experiments were performed in triplicate and the standard deviation was used as the measurement error. Bacteria inactivation percentages (Eq. [1]) were determined for a given condition according to the following:

$$\%\text{bacteria inactivated} = \frac{C_0 - C_n}{C_0}, \qquad (1)$$

where $C_0$ is the bacterial colony count of the control disc at time 0 (untreated sample after 20 min) and $C_n$ is the bacterial colony count after $n$ minutes of AquaROS spray, which is 20 min unless otherwise noted.

### Transmission electron microscopy (TEM) analysis

#### AquaROS-treated and untreated E. coli samples
Both AquaROS-treated (20-min spray of 5 µl *E. coli* in LB broth on stainless-steel disc in a spray chamber) and untreated (control) *E. coli* cells were centrifuged and the added LB broth was replaced with a fixative solution of glutaraldehyde and formaldehyde in phosphate-buffered saline for at least 1 h at room temperature. Multiple samples were sprayed and collected into one sample to ensure that enough bacterial cells were present for TEM sample preparation.

#### TEM sample preparation of AquaROS-treated and untreated E. coli
The cells were pelleted and re-suspended in 10% gelatin in 0.1 M sodium cacodylate buffer (pH 7.4) at 37 °C and allowed to equilibrate for 5 min followed by removal of excess gelatin and chilling in cold 1% osmium tetroxide for 2 h with rotation at 4 °C. After washing three times with cold ultra-filtered water, the cells were stained overnight in 1% uranyl acetate at 4 °C. The samples were dehydrated through a series of ethanol washes (30, 50, 70 and 95%) for 20 min each at 4 °C and finally at 100% ethanol twice followed by propylene oxide (PO) for 15 min. Samples were infiltrated into resin (Embed-812) mixed at ratios of 1:2, 1:1 and 2:1 with PO for 2 h each. Samples in 2:1 (resin:PO) were rotated at room temperature overnight. Samples were placed into resin for 2–4 h before placing into moulds with labels and fresh resin and placed at 65 °C overnight.

Sections (approximately 80 nm thickness) were picked up onto formvar/carbon-coated 100-mesh Cu grids followed by staining (1) for 30 s in 3.5% uranyl acetate in 50% acetone and (2) for 3 min in 0.2% lead citrate. TEM analyses were done with a JOEL JEM-1400 120-kV instrument (JOEL USA, Inc.. Peabody, MA, USA). Photos of the images were taken using a Gatan Orius 4K × 4K digital camera.

### Fragmentation of phospholipids mixture

#### Mass spectrometry analysis of the fragments of phosphatidylglycerol induced by AquaROS treatment
Phosphatidylglycerol (PG) solutions were prepared by dissolving 10 mM L-α-phosphatidyl-DL-glycerol molecules (Sigma-Aldrich) in water:ethanol (1:1, v/v). This solution was deposited onto polytetrafluoroethylene-printed glass slides with 5-mm diameter open wells. These wells were used to restrict the area of deposited PGs within the area of AquaROS spray treatment. The PG-deposited glass slides were dried in a desiccator for 10 min under vacuum. We sprayed AquaROS onto PG-deposited glass slides for 20 min and collected PGs in water:ethanol (1:1, v/v) solution.

#### Tandem mass spectrometry analysis
A high-resolution Orbitrap mass spectrometer (LTQ Orbitrap XL Hybrid Ion Trap Orbitrap; Thermo Scientific) was used for the mass spectrometry analysis. The identification of the observed fragmentation products resulting from AquaROS treatment was carried out by tandem mass spectrometry (MS/MS) using collision-induced dissociation (CID). To confirm the identities of the observed molecules, fragmentation patterns of fragmentation products were compared with standard samples that were acquired by CID or thermal fragmentation of PG molecules. Voltages at –5 kV and 44 V were applied to the electrospray ionization source and inlet capillary. The temperature of the heated capillary inlet was maintained at approximately 275 °C.

## Results and discussion
To test the efficacy of AquaROS at killing bacteria, we chose *E. coli*, a known faecal indicator and related to other enteropathogenic strains, and *S. typhimurium*, an enteric pathogen that causes food poisoning. Concentrations between $2 \times 10^8$ and $5 \times 10^8$ CFU ml$^{-1}$ of bacteria were inoculated (5 µl) onto stainless-steel discs. The number of surviving bacteria was measured after a 20-min exposure at room temperature to the microdroplet spray. Fig. 1 provides the schematic of the AquaROS generation device and the experimental setup. Bacteria were treated with water droplets having an average diameter of approximately 10 µm (Lee *et al.*, 2015; Nam *et al.*, 2017), which were produced by flowing chromatography-grade water (10 µl min$^{-1}$) through three fused-silica capillaries, each with a coaxial flow of nebulizing N$_2$ gas or air.

We investigated the effectiveness of the AquaROS treatment in killing *E. coli* and optimized the disinfection power of AquaROS by investigating different parameters of the spray system, including spray distance, gas pressure and water flow rate (Fig. 2). The effects of spray distance on killing *E. coli* was tested by varying the distance from 3 to 18 cm, while N$_2$ nebulizing gas (120 psi) and water flow rate (10 µl min$^{-1}$) were kept constant. The percentage of bacterial

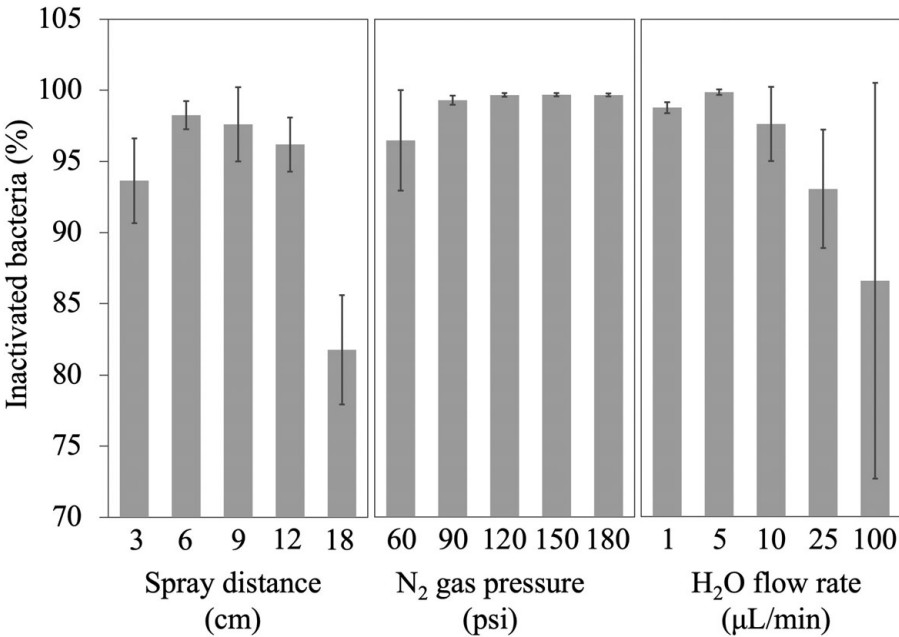

**Fig. 2.** Effects of spray distance, $N_2$ gas pressure and water flow rate on bacterial inactivation. Each bar represents three trials. For the spray distance experiments, $N_2$ gas pressure was 120 psi and water flow rate was 10 µl min$^{-1}$. For the $N_2$ gas pressure experiments, spray distance was 9 cm and water flow rate was 10 µl min$^{-1}$. For the water flow rate experiments, spray distance was 9 cm and $N_2$ gas pressure was 120 psi. After spraying, bacterial samples were serially diluted before plating onto LB agar plates by glass bead spreading method and incubated for 16–18 h at 37 °C.

inactivation was measured by comparing the cell count of *E. coli* treated with AquaROS to the ones with no treatment. Similar *E. coli* inactivation efficiency of ~97% at distances of 6, 9 and 12 cm was achieved. By comparison, less than 95% was achieved at 3 cm and only ~82% inactivation (with much larger error) was achieved at 18-cm spray distance. The lower inactivation efficiency at a short distance of 3 cm might be caused by a rapid dispersion of untreated bacteria out of the spray area after a few seconds of the microdroplet spray, resulting in the lack of sufficient exposure time to the spray. The significantly lower disinfection efficiency at a long distance of 18 cm is attributed to the ineffective delivery of ROS in microdroplets caused by the evaporation of water as well as the difficulty in aligning the spray to the sample. Although high inactivation was achieved at 6-cm spray distance, qualitative experiments showed dispersion or 'washing out' of *E. coli* colonies on LB agar plates during the spray at the short distance (Supplementary Fig. 1). Therefore, we determined the optimal distance to be 9 cm, which was used for the rest of the studies.

We tested the $N_2$ gas pressure dependence by spraying microdroplets onto *E. coli* on stainless-steel discs with different gas pressures, while the spray distance and the water flow rate were kept constant at 9 cm and 10 µl min$^{-1}$, respectively. We found that the inactivation efficacy of *E. coli* was near 100% for gas pressures between 90 and 180 psi, while inactivation was lower (~96%) at 60 psi (Fig. 2). We then measured the activity of AquaROS by varying the water flow rate at constant spray distance (9 cm) and $N_2$ gas pressure (120 psi). The inactivation efficacy of *E. coli* remained relatively constant under different flow rates ranging from 1 to 10 µl min$^{-1}$. However, significantly lower inactivation, <93%, was achieved at flow rates of 25 and 100 µl min$^{-1}$ because of dispersion of the bacterial cells from the spray area (Supplementary Fig. 2), suggesting that the water flow rate needs to be lower than 10 µl min$^{-1}$ to minimize dispersion. From these investigations,

we determined the optimal spray conditions to be 9 cm for spray distance, 10 µl min$^{-1}$ for water flow rate and 120 psi for nebulizing gas pressure.

To test the generality of AquaROS in killing bacteria, microdroplets were sprayed onto *E. coli* and *S. typhimurium* that were inoculated onto stainless-steel discs. The AquaROS spray for 20 min with nebulizing air operated between 90 and 115 psi killed over 98% of *E. coli* and *S. typhimurium*. Pressure switching between this pressure range did not affect the efficacy of AquaROS, as shown in Fig. 2.

Fig. 3 compares the inactivation efficacy of AquaROS towards wet and dry samples of *E. coli* and *S. typhimurium* on stainless-steel discs. The susceptibility of both bacteria to AquaROS treatment appeared to be similar. For *E. coli* and *S. typhimurium*, the inactivation efficiency for wet and dry is 99.99 and 99.39%, and 99.98 and 98.63%, respectively. The paired *t* test is *p* = 0.0024 (dry) and *p* < 0.0001 (wet), which validates the statistical significance of the data. Preliminary data suggest that as short as a 1-min spray is able to inactivate 96% of dried *E. coli*. Variations in the killing percentage might be caused by the biological variability of individual microorganisms in a population and slight variations in spray alignment with bacteria on the disc surface. Control experiments (water stream only and gas only) resulted in less than 10% inactivation of bacteria and 55% inactivation with 3% $H_2O_2$ (Supplementary Fig. 3), suggesting that the disinfection effect did not arise from the osmotic shock from water containing no salt or the mechanical pressure from nebulizing gas. To demonstrate the possible practical use of AquaROS, we also examined its action on spinach leaves (Supplementary Fig. 4) and found it to be quite effective (98.76% inactivation).

To shed light on the disinfection mechanism, we analysed AquaROS-treated *E. coli* using TEM. A role of the bacterial cell wall, which is composed of the outer membrane (OM), the

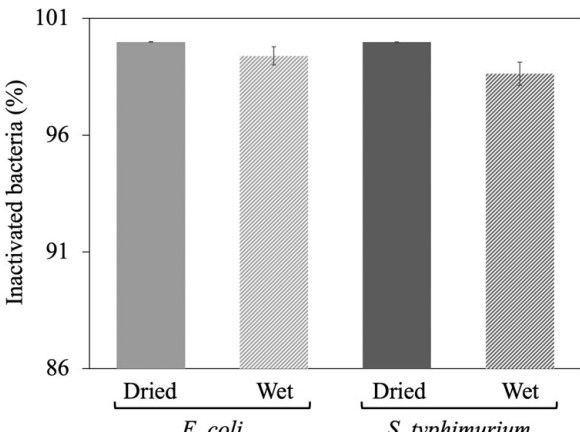

**Fig. 3.** AquaROS disinfection of *E. coli* and *S. typhimurium*. Wet samples were prepared by inoculating sterile stainless-steel discs with 5 µl bacteria sample. Dry samples were prepared in the same manner but followed with 5 min of drying under house vacuum. After spraying, serially diluted samples were spot plated (5 µl each) onto LB agar plates for *E. coli* and LB agar-streptomycin plates for *S. typhimurium*. Each bar represents one standard deviation from three replicates.

periplasmic space (PS) and the plasma membrane (PM), is to act as a protective barrier from its surrounding environment. Any disruption to this wall, whether temporal or permanent, can lead to cell death. TEM images (Fig. 4) revealed profound changes to the treated *E. coli* surface, notably morphological changes (Fig. 4*b*), OM damage and detachment (Fig. 4*c*), and OM blebbing ('bulging') of the outer membrane and compensatory shrinkage of the PS that results in leakage of the cytoplasmic contents (Fig. 4*d*) through a more permeable membrane. Blebbing occurs in the early phase of apoptosis and precedes the formation of apoptotic membrane protrusions that are shown in Fig. 4*d*. These membrane changes are in stark contrast to untreated (control) bacteria that appear structurally normal with the OM, PS and PM of the cell wall clearly defined and the rod-shaped morphology preserved (Fig. 4*a*). The average OM thickness of the untreated *E. coli* cells is $16.4 \pm 1.5$ nm ($n = 10$). The indistinct OM, PS and PM results in an average thickness of $25.1 \pm 3.3$ nm ($n = 9$) from the combined measurements of OM, PS and PM of the treated cells. In one of the TEM images (Supplementary Fig. 5), the OM of a treated cell with AquaROS was found to be $6.9 \pm 1.2$ nm, which is significantly reduced as compared with the untreated cell OM thickness. Furthermore, vacuolization of the cell as seen in Fig. 4*e* has a direct impact on the cell wall because it depletes the inner phospholipid membrane. Microtubules in the *E. coli* cell are seen in Fig. 4*f*, showing that these cells underwent apoptosis as microtubules play a crucial role in the reorganization of the cytoskeleton of bacteria during apoptosis.

These results suggest that the primary site at which AquaROS exerts microbicidal action is the cell wall through physical and chemical changes of the membrane components when exposed to the charged, high electric field surface of the water microdroplets and the ROS-mediated reactions involving ROS species present in the microdroplets upon impingement with the *E. coli* (Fig. 4) and *S. typhimurium* surfaces. These disruptions to the cell wall result in increased permeability to ROS and water, resulting in osmotic shock and leakage of the cell content (Pillet *et al.*, 2016).

The microdroplets formed in our spray have an associated electric field (at the air–water interface) on the order of $10^{10}$ V m$^{-1}$ (Kathmann *et al.*, 2008, 2009). This high electric field contributes to damage of the *E. coli* cell wall upon impingement of

the water microdroplets on the bacterial cell surface. Several studies on the deleterious effects of electric fields to the cell wall of bacteria have been reported. Exposure of the cell wall to an electric field leads to its physical damage and concomitant increase in its permeability, leading to cell death (Wang *et al.*, 2017). High-strength electric fields up to 20 kV cm$^{-1}$ at pulse durations of several milliseconds are shown to be incredibly lethal, killing at least 99.99% of bacterial cells because of damage to their cell membrane (Hülsheger and Niemann, 1980). Exposure of highly negatively charged microdroplets can induce high lateral stress from a change in the surface potential of the bacteria's cell membrane (Hülsheger and Niemann, 1980; Hülsheger *et al.*, 1981, 1983). Similarly, Unal and co-workers described a mechanism of cell death involving the dielectric breakdown of the cell membranes of bacteria, such as *E. coli* O157:H7 and *Listeria monocytogenes*, under the influence of a pulsed electric field, resulting in injury to the cell membrane (Unal *et al.*, 2002). The effects of charge at the surface of different particles on antimicrobial death have been reported (Abbaszadegan *et al.*, 2015; Salvioni *et al.*, 2017). Recently, charged nanodroplets of water with very strong surface charge of 10 electrons per droplet were produced and shown to kill bacteria on tomato surfaces (Pyrgiotakis *et al.*, 2015).

The same associated electric field at the surface of water microdroplets contributes to their interfacial reactivity, that is, the strength of the electric field ionizes OH$^-$ to form OH· (Lee *et al.*, 2019). OH$^-$, H$_2$O$_2$ molecules and OH· are identified ROS components present in water microdroplets. ROS-mediated chemical reactions also contribute to the inactivation of *E. coli* and *S. typhimurium* in our study. Cell membrane oxidation by ROS has been the subject of considerable attention because of its importance in causing cell wall destruction (Halliwell and Gutteridge, 2015; Wang *et al.*, 2017). Recently, we provided evidence for the presence of H$_2$O$_2$ in microdroplets (Lee *et al.*, 2019), which we believe is formed by the reaction of OH·. The role of ROS, such as H$_2$O$_2$, OH$^-$ and OH·, has been established in killing bacteria by oxidizing a diverse range of biological targets (DNA, lipids and proteins) in the cell's cytoplasm, disrupting cellular metabolic reactions leading to apoptosis (Linley *et al.*, 2012). In our experiments, ROS may play a role in killing *E. coli* and *S. typhimurium* during AquaROS spray. This idea is supported by Pyrgiotakis *et al.* (2015) who described inactivation of microorganisms with nanometre-sized engineered water nanostructures. The hydroxyl radical and singlet oxygen ($^1$O$_2$) are believed to be the most reactive in killing pathogens (Vatansever *et al.*, 2013). Although AquaROS contains H$_2$O$_2$, its concentration may be too low to play a significant role in the inactivation of bacteria. The concentration of H$_2$O$_2$ in microdroplets has been detected at 30 µM (Lee *et al.*, 2019). Typically, working concentrations of H$_2$O$_2$ for disinfection range from 188 mM to 8 M (0.4–30%) (Brudzynski *et al.*, 2011; Wang and Zhang, 2018). OH· can effectively damage the bacterial cell envelope given its high oxidation potential (2.80 V) (Finnegan *et al.*, 2010) and nonselectivity. Furthermore, OH· is more effective than H$_2$O$_2$ at oxidizing the cell envelope because of its higher oxidation potential (Pillet *et al.*, 2016).

To investigate the chemical mechanism of AquaROS, we mass spectrometrically analysed PG molecules treated with AquaROS. PG lipids were chosen because they are abundant in Gram-negative bacteria including *E. coli* (Sohlenkamp and Geiger, 2016). This mixture of PG lipids is a simple model of the complex OM of Gram-negative bacteria. Fig. 5*a* shows the spectrum of a mixture of untreated PG (18:1/16:0, **1**, *m/z* 747.52) and PG (18:1/18:0, **2**, *m/z* 755.55). Fig. 5*b* presents the products formed after AquaROS

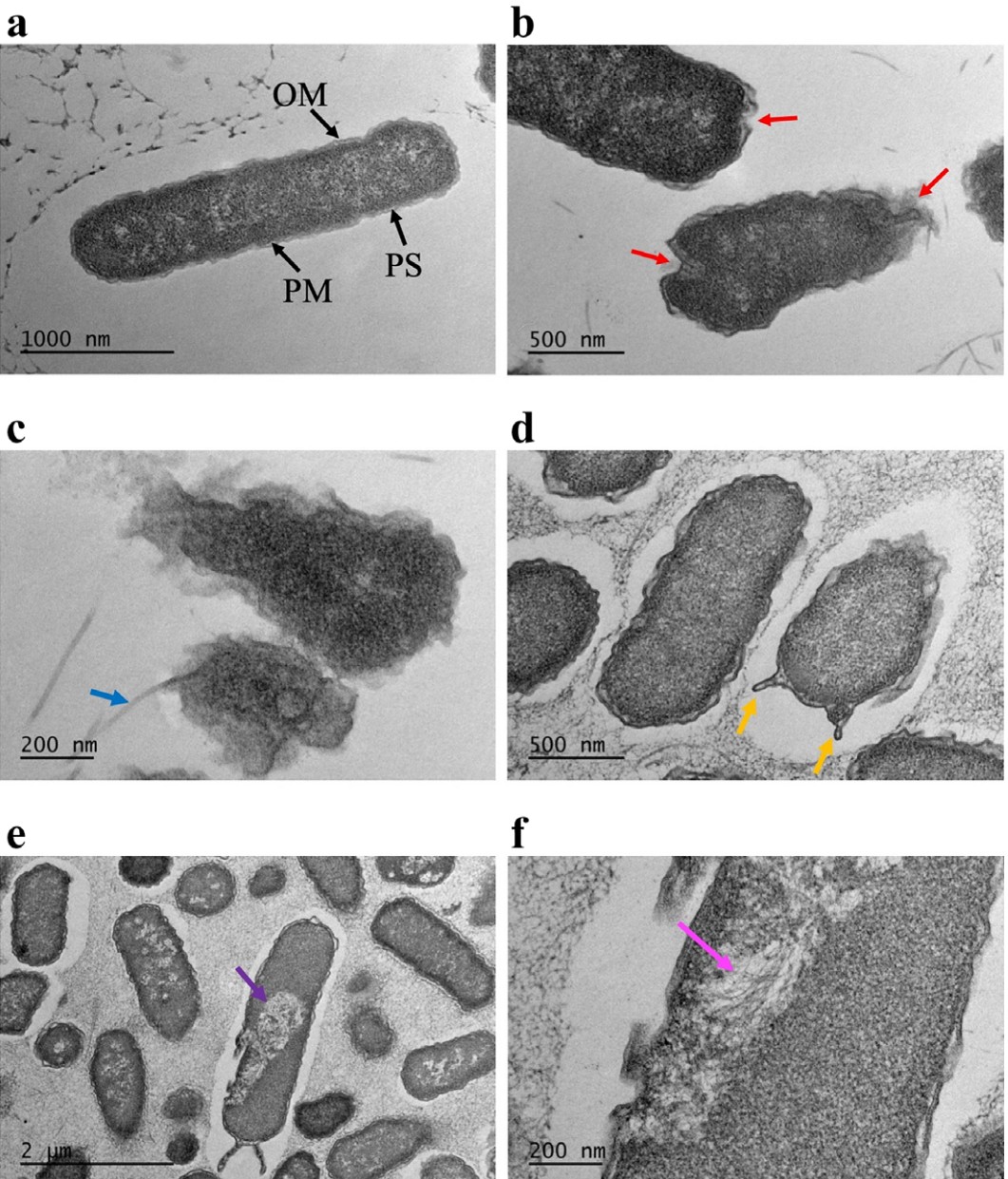

**Fig. 4.** TEM images of *E. coli* cells. (*a*) Control sample (no AquaROS spray). Arrows point to the outer membrane (OM), periplasmic space (PS) and plasma membrane (PM). Sample sprayed with AquaROS for 20 min. (*b*) Red arrows point to changes/damage to cell envelope's OM. (*c*) Blue arrow points to detached OM. (*d*) Orange arrows point to blebs in the OM. (*e*) Purple arrow points to large vacuole. (*f*) Magenta arrow points to area where microtubules are visible.

treatment of a mixture of **1** and **2**. These products are formed from the hydrolytic fragmentation at the C■O bond between the glycerol moiety and the carbon chains to form fragments **3** ($m/z$ 483.27) and **4** ($m/z$ 509.29), their identities were confirmed by tandem mass spectrometry (Supplementary Figs. 6 and 7, respectively). Control experiments ruled out drying, hydrolysis by water or mechanical effect in the formation of these fragments (Supplementary Fig. 8).

Free-radical ROS can directly alter the cell membrane by hydrogen abstraction from membrane-bound proteins and lipids to initiate a chain reaction to involve other lipids in the bilayer to form many lipid peroxides, resulting in the formation of fragmented species of oxidized lipids that can further react with proteins and lipids to form a complex population of products

(Abbaszadegan *et al.*, 2015; Salvioni *et al.*, 2017). These oxidized products can alter membrane permeability and also enter the cells to degrade cellular metabolism that can lead to cell death.

Another mechanism for the action of AquaROS on the cell membrane may likely involve the roles of interfacial chemistry between water and hydrophobic media occurring in nanobubbles. Perhaps the first study to recognize the importance of nanobubbles was one showing that degassed water is more effective than ordinary water in cleaning surfaces covered with hydrophobic materials, such as oil, in contact with water by enhancing the dispersion of hydrophobic particles in water (Pashley *et al.*, 2005). It was found that dissolved atmospheric gas molecules are drawn to the interface and assist the cavitation expected as two hydrophobic surfaces separate in water.

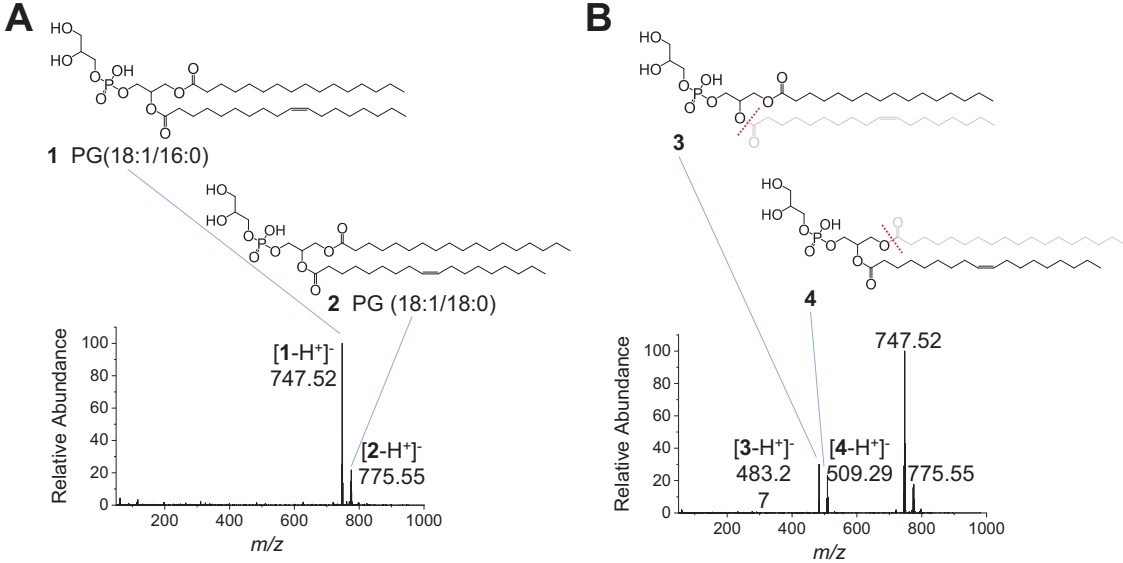

**Fig. 5.** Comparison of mass spectra of PGs **1** and **2**, which are present in *E. coli*, with intact PGs *versus* AquaROS-treated PGs. (*a*) Mass spectrum of standard sample (no AquaROS treatment). (*b*) Mass spectrum of AquaROS-treated PGs showing both fragmented structures **3** and **4** for PG **1** and PG **2**, respectively, and intact PG **1** and **2**.

This behaviour is consistent with the hypothesis that microscopic cavitation in a hydrophobic pocket might be the source of activation energy for cutting DNA by restriction enzymes in aqueous solutions (Kim *et al.*, 2001) Microscopic cavitation may also facilitate cell membrane rupture. Bubbling heated, unpressurized $CO_2$ through wastewater has been found to kill both bacteria and viruses, and this procedure can be scaled up for water treatment (Shahid *et al.*, 2014; Sanchis *et al.*, 2019). Moreover, the presence of small amounts of dissolved gases (Sanchis *et al.*, 2018) and electrolytes (Craig *et al.*, 1993) can have a profound effect on the creation of nanobubbles. It is possible that spontaneous nanobubble fluctuations at the air–water interface of microdroplets aid the formation of ROS. In addition, vibration of bulk water interfacing with air, which always accompanies spraying, can play an important role in promoting nanobubble formation (Fang *et al.*, 2020). These studies suggest that the disinfection with water microdroplets in the present study may share a similar mechanism as that with nanobubbles, and the disinfection efficiency of AquaROS can be improved by incorporating the formation of additional nanobubbles. However, many future experiments need to be carried out to discover those factors that might increase the potency of AquaROS. Further mechanistic studies are on-going, but the bactericidal effects of AquaROS are well demonstrated, we believe.

## Conclusion

Our study demonstrates the disinfection of *E. coli* and *S. typhimurium* with sprayed water microdroplets, which we call AquaROS achieves greater than 98% inactivation. We believe that the inactivation of bacteria involves chemical attack of ROS that are present in AquaROS on the bacterial cell wall, and its response to the external field of the charged microdroplets, resulting in damage to the cell wall. Because AquaROS is generated from ordinary water and ROS in the AquaROS are decomposed into water and oxygen, it shows promising effect as the direct application of a disinfectant

with a further benefit of it is residue-free, making this disinfectant environmental friendly.

**Open Peer Review.** To view the open peer review materials for this article, please visit http://doi.org/10.1017/qrd.2020.2.

**Supplementary Materials.** To view supplementary material for this article, please visit http://doi.org/10.1017/qrd.2020.2.

**Acknowledgements.** We are grateful to Lydia-Maria Joubert for useful discussions on the TEM data and to Carlos Aguilar for help with bacteria inactivation for TEM analysis.

**Author contributions** R.N.Z., M.T.D. and J.K.L. conceived of this study. M.T.D. and J.K.L. designed the research plan and interpreted experimental results. M.T.D. performed bacteria growth, colony counting and data processing with assistance from A.C.M. and R.N. All authors wrote this manuscript.

**Financial support.** This work was supported by the SPARK Program in Translational Research at Stanford grant (Title: Spontaneously Formed Reactive Oxygen Species from Water for Pathogen Disinfection). TEM analysis was supported, in part, by ARRA, Award Number 1S10RR026780-01, from the National Center for Research Resources (NCRR).

**Conflict of interest.** The authors declare no conflict of interest.

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
