## [Reviewer Report]

*Comments to Author*: Spraying Small Water Droplets Acts as a Bacteriocide. QRD-D-20-00002

I think this is an excellent sample of submission fitting QRB-Discovery, the surprising (though not fully explainable) observation that air-nebulized water micro-droplets have a clear disinfectant effect - controls with bulk water and air showing no effect. Reactive oxygen species are speculated to be responsible, possibly in combination with droplet surface charge. Effects of electric fields, free-radical oxygen species and hydrogen abstraction from membrane-bound proteins are also discussed as possible origins but the mechanism(s) appear to remain enigmatic.

The ms reads well and I suggest it be accepted for publication in QRB-D.

Minor comment.

Over the years many ‘physical’ phenomena have been proposed in disinfection contexts. I suggest two additional examples be referred to from the Ninham group: that spontaneous bubble fluctuation in water at hydrophobic surface may create radicals and the observation that carbon dioxide bubbles in water have inactivation effect on virus and bacteria, as these effects might be related to those reported here (bubble and droplet curvature and electric fields?).

H.-K. Kim, E. Tuite, B. Nordén and B. W. Ninham

Co-ion dependence of DNA nuclease activity suggests hydrophobic cavitation as a potential source of activation energy. European Physical Journal E: Soft Matter, (2001), 4 (4), 411-417. DOI: 10.1007/s101890170096

Garrido Sanchis, A., R. Pashley, and B. Ninham, Virus and bacteria inactivation by CO2 bubbles in solution. NATURE partner journal Clean Water, 2019. 2(1): p. 5. https://doi.org/10.1038/s41545-018-0027-5

---

## [Reviewer Report]

*Comments to Editor*: Paper acceptable after authors’ response and possible minor revision Bengt

*Comments to Author*: Reviewer #1: Spraying Small Water Droplets Acts as a Bacteriocide. QRD-D-20-00002

I think this is an excellent sample of submission fitting QRB-Discovery, the surprising (though not fully explainable) observation that air-nebulized water micro-droplets have a clear disinfectant effect - controls with bulk water and air showing no effect. Reactive oxygen species are speculated to be responsible, possibly in combination with droplet surface charge. Effects of electric fields, free-radical oxygen species and hydrogen abstraction from membrane-bound proteins are also discussed as possible origins but the mechanism(s) appear to remain enigmatic.

The ms reads well and I suggest it be accepted for publication in QRB-D.

Minor comment.

Over the years many ‘physical’ phenomena have been proposed in disinfection contexts. I suggest two additional examples be referred to from the Ninham group: that spontaneous bubble fluctuation in water at hydrophobic surface may create radicals and the observation that carbon dioxide bubbles in water have inactivation effect on virus and bacteria, as these effects might be related to those reported here (bubble and droplet curvature and electric fields?).

H.-K. Kim, E. Tuite, B. Nordén and B. W. Ninham

Co-ion dependence of DNA nuclease activity suggests hydrophobic cavitation as a potential source of activation energy. European Physical Journal E: Soft Matter, (2001), 4 (4), 411-417. DOI: 10.1007/s101890170096

Garrido Sanchis, A., R. Pashley, and B. Ninham, Virus and bacteria inactivation by CO2 bubbles in solution. NATURE partner journal Clean Water, 2019. 2(1): p. 5. https://doi.org/10.1038/s41545-018-0027-5